# The Role of HERV-K in Cancer Stemness

**DOI:** 10.3390/v14092019

**Published:** 2022-09-12

**Authors:** Sarah R. Rivas, Mynor J. Mendez Valdez, Vaidya Govindarajan, Deepa Seetharam, Tara T. Doucet-O’Hare, John D. Heiss, Ashish H. Shah

**Affiliations:** 1Surgical Neurology Branch, National Institute of Neurological Diseases and Stroke, Bethesda, MD 20892, USA; 2Section of Virology and Immunotherapy, Department of Neurosurgery, Leonard M. Miller School of Medicine, University of Miami, Miami, FL 33136, USA; 3Neuro-Oncology Branch, Center for Cancer Research, National Cancer Institute, Bethesda, MD 20892, USA

**Keywords:** human endogenous retroviruses, HERV-K, stem cell phenotypes, carcinogenesis

## Abstract

Human endogenous retrovirus-K (HERV-K) is the most recently integrated retrovirus in the human genome, with implications for multiple disorders, including cancer. Although typically transcriptionally silenced in normal adult cells, dysregulation of HERV-K (HML-2) elements has been observed in cancer, including breast, germ cell tumors, pancreatic, melanoma, and brain cancer. While multiple methods of carcinogenesis have been proposed, here we discuss the role of HERV-K (HML-2) in the promotion and maintenance of the stem-cell in cancer. Aberrant expression of HERV-K has been shown to promote expression of stem cell markers and promote dedifferentiation. In this review, we discuss HERV-K (HML-2) as a potential therapeutic target based on evidence that some tumors depend on the expression of its proteins for survival.

## 1. Introduction

Human endogenous retroviruses are remnants of retroviral germline infections that were incorporated into the human genome several million years ago. It is estimated that 8% of the current human genome consists of endogenous retroviral elements. However, most of these HERV elements have been altered by mutations, insertions, and deletions that render them inactive [1]. HERV-K (HML-2), first reported in 1986, is the most recently integrated retrovirus which contains a near-full length transcript in the human genome [2]. Although generally dormant, open reading frames of HERV-K genes *gag* (group-specific antigen), *pol* (polymerase), and *env* (envelope) can be read and translated into functional retroviral proteins [3,4,5]. The role of HERV-K (HML-2) in gene regulation, embryonic development, pathogenesis, and cancer has been widely studied. In this review, we discuss HERV-K’s involvement in pluripotency, self-renewal capacity, dedifferentiation, and other markers of cancer stemness. Additionally, we surveyed the literature to identify functional studies from 1986 to present that explored the expression of the HERV-K gene, mRNA, or protein in malignancy.

## 2. HERV-K Promotes Carcinogenesis

The relationship between HERV-K (HML-2) and human disorders has been well established [6,7,8] with increased emphasis on cancer in recent years [9,10]. HERV-K loci are actively transcribed during embryogenesis and transcriptionally repressed post-development in healthy human tissues [11]. HERV-K overexpression is correlated with malignant phenotypes and upregulated in multiple cancers, including breast [12,13], lymphoma [14], germ-line tumors [15], and melanoma (Table 1) [16]. Previously, studies have shown that HERV-K overexpression correlates with malignant phenotypes. For example, a large proportion of breast cancer tissues (~45%) express increased HERV-K RNA transcripts compared to normal tissue controls [12,13]. Similarly, in low grade ovarian cancer, HERV-K env protein is expressed on the cell surface but not in normal ovarian epithelial tissue [17]. Table 1 discusses functional studies that have outlined expression of various HERV-K elements in cancer.

Much discussion surrounds HML-2 as an oncogenic driver, but two major mechanisms have been proposed. First, is that dysregulation of gene expression can increase the likelihood of tumorigenesis, by HERV-K long terminal repeat (LTR) cis-activation of host gene expression [29,30]. The 5′ LTR, which is the known promoter of HML-2 element and regulates its expression, was discovered to be fused to the oncogene ETV1 in prostate cancer cell lines, promoting ETV1 expression and cancer cell invasion [31,32]. Second, is that aberrant expression of functional HML-2 proteins alters the regulation of oncogenic pathways [18,33]. In breast cancer, Zhou et al. demonstrated that HERV-K env protein expression was essential for cell proliferation, migration, and invasion [13,18,33]. Specifically, they demonstrated that abrogation of env protein expression decreased the presence of epithelial-mesenchymal transition (EMT) markers and attenuated tumor progression [18]. HERV-K envelope splice products such as Rec and Np9 are expressed in normal tissues, but their physiologic role is clearly defined [34]. However, Np9 and Rec have been shown to regulate viability, migration, and invasion [35]. Chan et al. sensitized teratocarcinoma cell lines to chemotherapeutic agents by depleting cells of Np9 and showed that Np9 inhibition decreased migration [15]. Furthermore, Np9 RNA expression is upregulated in EBV-transformed lymphomas, as well as leukemia and melanomas [36,37,38]. Furthermore, HERV-K env protein overexpression promotes malignant phenotypes through upregulation of the RAS/ERK/MEK in meningioma, schwannoma, and pancreatic cancers [22,23]. Examples of these proposed mechanisms are shown in Figure 1.

## 3. Classical Markers of Stemness

HERV-K and its protein products have been demonstrated as potential markers of stemness since they are involved in cancer cell self-renewal capacity and pluripotency. For example, Mareschi et al. previously noted a significant correlation between HERV-K and mesenchymal stem cell expression of several stem cell markers, including nanog homeobox (NANOG), octamer binding transcription factor 4 (OCT-4), and SRY-Box Transcription Factor 2 (SOX-2), all of which have demonstrated roles in maintaining stemness. [39]. Most importantly, these genes have notable roles in oncogenesis, as they have been associated with the reprogramming of adult somatic cells into pluripotent stem cells [40,41].

OCT-4 is a transcription factor that regulates target gene expression by binding to the octamer sequence 5′-ATGCAAAT-3′ located in the promoter and enhancer regions of target genes [42]. Physiologically, OCT-4 plays a key role in maintaining stemness during mammalian embryogenesis [43]. In oncogenesis, OCT-4 appears to drive dedifferentiation of somatic tissues, possibly by inducing the expression of other genes associated with stemness, thus reprogramming somatic cells into ones reflecting a more stem-like phenotype [44]. 

NANOG is a homeobox gene that has been well defined as a driver of cancer stemness. Classically, NANOG and other homeobox genes are critical in body segmentation and cellular differentiation [45]. These genes regulate gene expression via internal 180 base pair sequences known as homeodomains, which regionally bind other gene loci, thus acting as transcription factors [46]. Similar to OCT-4, NANOG also plays a role in embryogenesis, but has been shown to potentially be under the control of OCT-4 and other markers of stemness [47].

SOX-2 is the most well-known of the SOX family of genes. SOX-2, like other members of its genetic family, possesses a key DNA-binding element which classifies it as a transcription factor [48]. Similar to NANOG, SOX-2 typically associates with other transcription factors, like OCT-4, to influence gene transcription [49]. SOX-2, in conjunction with OCT-4, C-MYC, and other stem cell related transcription factors, is used to reprogram differentiated cells into induced pluripotent stem cells [49].

There are several in vitro models that can characterize the role of HERV-K in stemness exhibited in both cancer cell lines and patient-derived cancer samples. Neurosphere generation and subsequent assay for markers of stemness are well described in the literature [50]. Multiple groups have used cell line or patient-derived neurospheres for downstream analysis of novel markers of stemness such as HERV-K [28]. Bazzoli et al. discovered that myeloid Elf-1 like factor (MEF) preserved stemness in cultured neurospheres [51].

## 4. Mechanisms of HERV-K and Stem Cell Expression

Although much of the physiological role of HERV-K has yet to be understood, multiple groups have found that its expression contributes to undifferentiated phenotypes and the maintenance of that phenotype [52]. HML-2 is found to be transcriptionally active during embryogenesis. HML-2 correlates with expression of pluripotency markers such as NANOG, OCT-4, and SOX-2, while silencing of HML-2 occurs during differentiation [31,39,52]. In neuronal differentiation, HML-2 activation increased the expression of neurotrophic tyrosine receptor kinase 3 (NRTK3), a membrane-bound receptor that when activated decreases microtubule-associated protein 2 (MAP2) expression [53]. MAP2 expression is low in neuronal precursors and is mainly expressed in neurons for stabilizing microtubules in neuronal axons and dendrites [54,55]. Therefore, HERV-K (HML-2) hyperactivation impaired human embryonic stem cell differentiation, and its over expression in cortical neurons impairs their activity [53].

In addition, HML-2 has been associated with cell-to-cell adhesion and differentiation through regulation of the mammalian target of rapamycin (mTOR) signaling [11]. Env protein was discovered to associate with 4F2 cell surface antigen heavy chain (CD98HC), a heterodimeric amino acid transporter known for regulating stem cell morphology in pluripotent stem cells via mTOR signaling [56]. HERV-K triggers CD98HC mediation of the mTOR pathway, which in turn regulates gene expression through LPCAT1. Wang et al. found that inhibition of HML-2 env protein resulted in decreased self-renewal capacity, decreased OCT-4 expression, and induced differentiation [11]. Further research has shown that not only does HERV-K regulate pluripotency factors, but stem cell regulatory factors mediate HERV expression in development. For example, LTR5(HS) retains the OCT-4 binding domain, and when bound, OCT-4 positively regulates the expression of the HERV-K element [30,57,58]. In addition, this group showed that DNA hypomethylation and OCT-4 binding to LTR5(HS) synergistically facilitate transcription of HERV-K [58].

Additionally, HERV-K contributes to the pluripotent features observed in atypical teratoid rhabdoid (AT/RT) tumors [28]. AT/RT is characterized by the inactivation of SMARCB1, a subunit of the BAF chromatin remodeling complex, which is necessary for neuronal differentiation [59]. Under normal expression of SMARCB1, HERV-K HML-2 expression is tightly controlled and transcription is low. However, when SMARCB1 expression is lost, HERV-K (HML-2) expression increases. The authors revealed that SMARCB1 loss resulted in increased C-MYC binding to the HML-2 LTR, resulting in its robust expression in this embryonal tumor. In this study, the authors observed HERV-K env expression promoted cell proliferation and a stem cell like phenotype via the NRAS pathway [28]. Figure 2 highlights HML-2’s involvement in normal neuronal differentiation and in oncogenic pathways.

## 5. Role of HERV-K and Preservation of Stemness

Mechanistically, dysregulation of critical genes in stemness regulation follows similar patterns to HERV-mediated dysregulation. Physiological hypomethylation of HERV-K, for example, correlates with upregulation of OCT-4 [58]. Similarly, hypomethylation at the HERV-K locus may preserve stemness in cancer cells and drive tumor invasiveness.

Recently, many publications have revealed increasingly intricate mechanisms by which HERV-K expression promotes stem cell-like features. For example, Argaw-Denboba et al. demonstrated expansion of CD133+ melanoma, characterized by preserved stemness, with concomitant overexpression of HERV-K env [60]. The authors noted that culturing melanoma cells in stem cell media increased their self-renewing capacity, migration, invasiveness, and expression of CD133 and HERV-K env [60]. The authors were also able to demonstrate a reduction in CD133 subtype expansion in response to knockdown of HERV-K transcripts.

## 6. HERV-K (HML-2) as a Target in Cancer

As previously stated, the requirement for HERV-K expression in multiple cancers has been well established, suggesting its candidacy as a therapeutic target [18,23,28]. HERV-K env protein expression in breast cancer has been correlated with disease progression, disease stage, lymph node metastasis, and reduced overall survival [61]. In the basal subtype of invasive ductal carcinoma (IDC), HERV-K env expression was significantly upregulated when compared to other subtypes [62]. Interestingly, basal cell is one of the most aggressive subtypes of triple-negative breast cancer (TNBC) with no effective targeted therapy. HERV-K env expression in patients with hepatocellular carcinoma (HCC) has been shown to serve as an independent prognostic indicator of overall survival and has been associated with cirrhosis, tumor differentiation, and staging [26]. Some evidence also exists for HERV-K expression in patients with Hodgkin’s Lymphoma (HL) with a significant drop in titer levels after appropriate treatment [14].

Given the significant upregulation of HML-2 in various cancers, it may serve as a potential therapeutic target, yet how to take advantage of these elements as targets in cancer remains to be explored. According to one method, HML-2 aberrant expression makes them a novel target for immune-mediated therapy. Several in vitro studies have shown promising results targeting HML-2 proteins. A study by Wang-Johanning et al. showed apoptosis and growth inhibition in breast cancer cell lines treated with anti-HERV-K monoclonal antibodies (mAbs), and a murine model demonstrated reduced growth of xenograft tumors in mice treated with the same mAbs [63]. Similarly, an in vitro study conducted on mononuclear cells from breast cancer patients showed a significant reduction in tumor growth and HERV-K mediated cytotoxicity on cells treated with HERV-K env protein chimeric antigen receptor (K-CAR) T-cells. An in vitro study by Krishnamurty et al. generated HERV-K env specific CAR T cells after demonstrating upregulated expression of HERV-K env protein in melanoma cells but no expression in non-malignant cells of the same patients [24]. Further, antigen-mediated tumor lysis of HERV-K env + cells by CAR-T cells reduced tumor burden in mouse models. These results suggest that HERV-K env protein may be a suitable therapeutic target, as in vitro models demonstrated elevated expression and in vivo models showed a reduction in tumor burden after treatment. Recently, Bonaventura et al. showed that tumors can present HERV epitopes on HLA-A2 molecules which can be identified and targeted by the immune system. HERV-epitope specific CD8+ T cells showed specificity to HERV-epitope presenting MDA-MB-231 cells in vitro and a significant increase in apoptosis in tumor cells and IFN-y production compared to negative control T cells. Additionally, they found HERV-specific tumor infiltrating lymphocytes in triple-negative breast cancer patient samples [64]. Finally, recent drug repurposing studies have assessed antiretroviral antitumor capability in high HERV-K expressing tumors. Retroviral protease inhibitors ritonavir, atazanavir, and lopinavir have been tested against Merlin-negative grade I meningioma cells. Such treatment decreased proliferation, inhibited HERV-gag maturation, and decreased HERV-K env protein expression [23]. Treatments targeting HERVs in malignancy are outlined in Table 2. 

Given the significant upregulation of HERV-K in several malignancies and preliminary evidence demonstrating tumor response to HERV-K mediated epigenetic treatment therapies, targeting HERV-K continues to be a promising avenue for research. Further characterization of HERV-K expression is necessary as not all malignancies express HML-2. More specific targeting of HML-2 may lead to more effective therapies and potentially stronger anti-tumor effects. Cellular HML-2 knock-out (KO) models may provide further insight into oncogenic development as it pertains to epigenetic dysregulation, however the role of HERV-K HML-2 in the normal development must be considered. Similarly, therapies targeting HML-2 in early stages of development, such as vaccination, may impair normal development, so when targeting HML-2 as a preventative measure, care must be taken. Early studies have shown HERV-K HML-2 to be a reasonable target for various cancers, but further studies are needed to establish its efficacy in in vivo models.

## Figures and Tables

**Figure 1 viruses-14-02019-f001:**
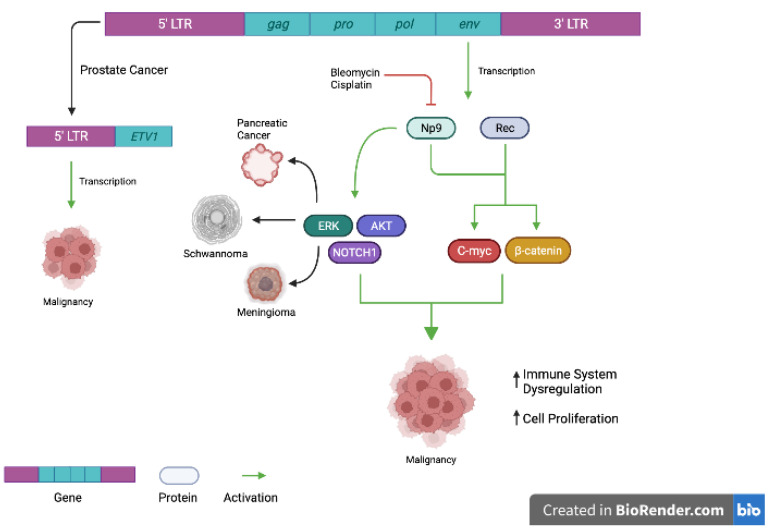
Examples of proposed mechanisms of HERV-K (HML2) oncogenesis. In prostate cancer, the 5′ long terminal repeat (LTR) may be fused to the ETV1 oncogene. Transcription of this gene may lead to impaired cell cycle regulation and malignancy. Similarly, transcription of HERV-K (HML2) env protein may lead to upregulation of Rec and Np9 proteins, which have been shown to be oncogenic drivers of disease. Through various downstream signaling pathways, they may increase immune system dysregulation and cell cycle proliferation, promoting oncogenesis. Figure 1 was made with Biorender.com (accessed on 29 June 2022).

**Figure 2 viruses-14-02019-f002:**
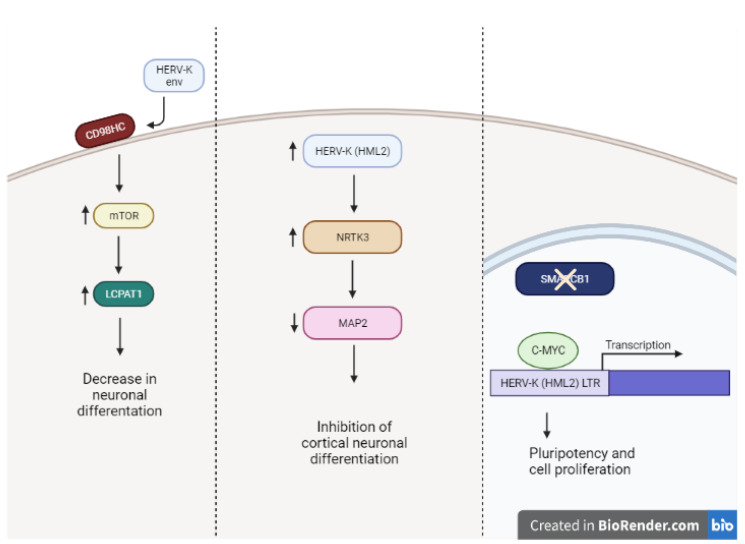
Proposed mechanisms of HERV-K expression and stemness in neurons. Exogenous HERV-K env mediates stemness by binding to CD98HC, which regulates mTOR-LPCAT1 pathway. Additionally, increased HERV-K (HML-2) expression increases NRTK3, after which a decrease in neuronal marker MAP2 is observed, promoting an undifferentiated state. Finally, in AT/RT cells, loss of transcription factor SMARCB1 allows binding of C-MYC to HERV-K LTR, promoting features characteristic of pluripotency and proliferation. Made with Biorendr.com (accessed on 29 June 2022).

**Table 1 viruses-14-02019-t001:** Expression of HERV-K elements in cancer.

Reference	Cancer	HERV-K Element	Main Finding
Wang-Johanning et al., 2001 [13]	Breast	HERV-K	Authors found HERV-K transcripts in breast cancer cell lines and tumor tissue but absent in non-malignant breast tissue.
Zhou et al., 2016 [18]		HERV-K env protein	HERV-K env protein is an essential element for tumorigenesis.
Wang-Johanning et al., 2007 [17]	Ovarian	HERV-K env protein	Surface expression of HERV-K env proteins was observed in ovarian cell lines and tissues. Additionally, anti-HERV antibodies were detected in patient samples.
Kleiman et al., 2004 [19]	Germ-cell tumors	anti-HERV-K antibodies	Authors detected anti-HERV-K antibodies in 67% of patients. They found serological antibody levels indicative of disease progression.
Sauter et al. 1995 [20]	Seminoma	HERV-K10 gag protein	Gag proteins were expressed in and secreted from Tera 1 cells. Seminoma also had elevated gag proteins. Patients exhibited high HERV-K antibody titers.
Chan et al., 2019 [15]	Teratocarcinoma	Np9	Np9 inhibition of NCCIT cells sensitized them to chemotherapeutics and environmental stresses.
Dolci et al., 2020 [21]	Colon	HERV-K (HML-2) HERV-K env proteins	Fifty-eight patient samples demonstrated a mean 61.56% decrease in HERV-K methylation. Env protein was expressed in tumor tissues but not in surrounding normal cells.
Li et al., 2017 [22]	Pancreatic	HERV-K env transcripts	Authors detected type 1 and type 2 HERV-K env transcripts in multiple pancreatic cell lines.
Maze et al., 2022 [23]	Meningioma Schwannoma	HERV-K env, gag Rec Np9	IHC staining was positive for intracellular expression and surface expression of HERV-K env protein on Sch-*NF2^+/+^* cells.
Krishnamurthy et al., 2015 [24]	Melanoma	HERV-K env protein	IHC staining of primary cells and metastatic melanoma cells showed over a 200-fold increase in HERV-K env expression.
Contreras-Galindo et al., 2008 [14]	Lymphoma	HERV-K RNA HERV-K env and gag proteins	Patients with lymphoma presented with higher titers of HERV-K RNA, env, and gag proteins in plasma.
Depil et al., 2002 [25]	Leukemia	HERV-K gag gene	Transcription of the HERV-K gag gene in leukemia blood samples was ten times normal.
Ma et al., 2016 [26]	Hepatocellular carcinoma	HERV-K	Results showed increased HERV-K expression in HCC tumor samples. HERV-K proved to be a negative prognostic marker.
Yuan et al., 2021 [27]	Glioblastoma	HERV LTR	GBM showed upregulation of LTR5_LTR-ERVK expression compared to normal cells or tissues.
Doucet-O’Hare et al., 2021 [28]	Atypical teratoid rhabdoid tumors	HERV-K env	HERV-K (HML-2) env protein expression was observed in AT/RT cell lines and 95% of patient tissue samples.

Hepatocellular carcinoma, HCC; Human endogenous retrovirus-K, HERV-K.

**Table 2 viruses-14-02019-t002:** Therapeutic approaches for targeting HERV-K in cancer.

Reference	Cancer	HERV Element Targeted	Method
Krishnamurthy et al., 2015 [24]	In vitro: melanoma cell lines, A888, A624, A375, and A375-SM In vivo: NSG mice injected with A375-SM-RmK cells	HERV-K env protein	HERV-K env-specific CAR^+^ T cells were generated by the SB system and expanded on AaPC. They were then introduced to cell lines or infused into a mouse model.
Wang-Johanning et al., 2012 [63]	In vitro: Breast cancer cell lines MCF-7, MDA-MB-231, and SKBR3. In vivo: NCr-nu/nu mice injected with MDA-MB-231 cells	HERV-K	Anti-HERV-K monoclonal antibodies were introduced to cell lines or injected into mouse models.
Zhou et al., 2015 [65]	In vivo: xenograft models injected with MDA-MB-231 or MDA-MB-435.eB1 cells	HERV-K env	Authors generated CAR T-cells against the HERV-K env protein using a mouse monoclonal antibody.
Bonaventura et al., 2022 [64]	In vitro: targeted HERV-epitope presenting MDA-MB-231 cell.	HERV	Used bioinformatics to identify epitopes homologous to cancer-associated HERV-epitopes. Generated CD8+ T cells specific to HERV epitopes to assess tumor-killing capability.
Maze et al., 2022 [23]	In vitro: MN-GI-*NF2^−/−^* cells	Retroviral protease	Cell viability was assessed after the retroviral protease inhibitors ritonavir, atazanavir, and lopinavir were administered

DNA methyltransferase inhibitors, DNMTi; Histon deacetylase inhibitors, HDAC; Sleeping Beauty, SB; Artificial activating and propagating cells, AaPC; Non-small cell lung cancer, NSCLC.

## Data Availability

Not applicable.

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
