# Peer review of "The Role of HERV-K in Cancer Stemness"

_viruses, 2022, doi:10.3390/v14092019_

Round 1
Reviewer 1 Report
Review Rivas et al.
The review, “The role of HERV-K in Cancer stemness” is a timely review following important publications linking HERV-K to single oncogeneic driver mutations and cancer stem cell phenotypes. Its shortcomings is that it stays relatively superficial, reviewing association and for the most part staying out of consideration of what comes first, eg. is HERV-K induced by stem cell markers or driving the stem cell markers and their phenotypes. The evidence is not always there, but this discussion should be part of a mechanistic review.
The figure 1 attempts to provide a mechanism for a role for HERV-K in cancers, but completely ignore the studies, including several cited here, which points to the expressed surface env protein as a driver and target for biological effects in cancer. This will have to be improved. The lack of focus on Env is perplexing, because the Env protein is not ignored in stemness, but there is no attempt to link these findings to cancers in general which seems a central point of the review. When there is a postulated mechanism for stemness in figure 2, it is exclusively based on neurons, including 2 non-oncogenic context and not discussed in the text under the brief section claiming to discuss cancer mechanisms. Key evidence has also been left out in Lemaitre et al. where HERV-K env seem to directly contribute to oncogenic pathway activation, a mechanism linked to the TM domain which is not an obvious binding partner in the CD98hc binding assays performed by Avindra Nath’s team. If the review will claim to review the role of HERV-K in cancer stemness, these kinds of interactions should be thoroughly reviewed and discussed.
In summary, the review requires some more work before it does full justice to the topic that richly deserves a timely review.
Comments through the text:
Abstract, HERV-k, K not capitalized
“some tumors depend on expression of its proteins for survival”, this is a key statement that is assayed in several cited manuscripts, including the recent single driver mutation driven cancer (Doucet-O’Hare & Maze), but only mentioned directly once in the table for a different study.
Reference 16, Feng Wang-Johanning et al. In this publication surface expression was not measured, but it was done in later studies.
Table 1: It is unclear what selection criteria has been used for including manuscript in the table. This should be clear and then the table should be exhaustive under the applied selection criteria. It would also be beneficial to systematically mark if functional studies have been done and the nature of the expression analysis performed Currently this is inconsistent.
Reference 17: This timepoint would be a good place to cite Lemaitre et al. as their measurement of signaling after controlled overexpression better fits the claim. The cited study is one of several similar ones, and could in theory have its results due to other expression effects.
Reference 25: Please state of HERV-K is a positive or negative prognosticator
Reference 23: The section referencing Krishnamurty et al. is a mess with the citation in the middle and no structure in going from in vivo to in vitro or the other way around. Also, please explain the significance of harvesting the PBMC’s from cancer patients as it is cited, when they are used to generate CAR-T’s to treat a cell line in immune deficient mice.
Table 2: The table has unclear inclusion criteria and should under this heading include studies of ALS, multiple sclerosis and diabetes treatment.
It could read ….targeting HERV-K in human cancer cells, but it would be better for the way it is used in the text to include Barbara Schnierle and her teams work with MVA vectored HERV-K Gag and Env vaccines.
The most important omission is probably Sacha et al which immunized NHPs against HERV-K, but see below
Concluding paragraph:
“may of HML-2” should read “of HML-2 may”
The last 7 lines appears quite confusing throughout. Initially I thought about cellular KO models and not mouse models as I assume the authors are aware that there are no direct homologues of human ERV-K’s in rodents. IAPE’s are distantly related, but it is currently unclear if they play a similar role.
Targeting HML-2 in early stages, what does this mean? From the context of development, tt seems to apply targeting HERV-K in young children or in utero. That would certainly leave some cancers available for targeting
When mentioning side effects in in vivo models, the lack of citation of Sacha et al. has to be corrected and discussed. This was a thorough attempt to provoke simian ERV-K specific immune pathology. This finding is one important aspect of the expected side-effects of ERV-K immunotherapies and targeting of this stem cell marker, but it have to be taken into account that this is not the first stem cell marker targeted in drug development (indeed fetal tissues have been experimentally used as immunogens for cancer prophylaxis) while another could be taken from a discussion on side effect profiles in cancer dominated by neoantigen and ERV responses (eg. melanoma and renal clear cell carcinoma, there are no obvious differences).
I think that if the side effect profile in relation to targeting HERV-K should be discussed, it should be so with relevant references and in a relevant context.
Reviewer 2 Report
Please see attachment.

Reviewer 3 Report
In the present manuscript entitled “The role of HERV-K in Cancer Stemness”, Rivas and co-authors reviewed the state of the art on the topic of endogenous retroviruses, specifically HERV-K, in cancer stemness. The topic is of great interest and it is worth publishing reviews and articles in this field. Given the increasing interest in endogenous retroviruses, a number of reviews have recently been published on the topics “HERVs and disease” and “HERVs and cancer”, and, unfortunately, not always by experts in the fields, and frequently not innovative. The authors of the present manuscript have the expertise and have already published on the topic. Although the present review has a similar structure to other reviews on HERVs role in cancer, this reports an update of the most recent works and is dedicated also to the specific aspect of “stemness”.
Specific aspects to be addressed:
Much has been published in the field of HERVs in tumours or HERVs in cell identity and embryonic development, but very few are related to HERVs and cancer stemness. The authors focused more on the first two aspects, but the section on the HERV role in cancer stemness should be improved. The authors correctly mentioned in the references some milestone reviews on the topic of HERVs and Cancer, but they lack to mention the first and comprehensive review published on the specific topic (Matteucci et al. Seminar in Cancer Biology 2018). There are also few papers reporting modulation of embryonic genes in cancer cells in association with HERV expression, still some articles are missing in the present manuscripts. Moreover, cancer stemness is peculiarly linked to cell plasticity, and articles that report the role of HERVs in those features have been published and are still missing in this manuscript.
Round 2
Reviewer 1 Report
After the initial comments, I raised requests for more elaborate content in tables and mechanistic figures as the titles referrals suggested a comprehensive content. The authors have chosen to narrow the definitions instead. This off course means that the claims is justified by the content, and makes it acceptable, but also leaves room for improvement.
Reviewer 3 Report
The authors clarified most of the concerns raised in the first round of review.